# Exploring the factors associated with the mental health of frontline healthcare workers during the COVID-19 pandemic in Cyprus

**Konstantinos Kapetanos**[1]*, **Stella Mazeri**[2], **Despo Constantinou**[3,4], **Anna Vavlitou**[5], **Marios Karaiskakis**[6], **Demetra Kourouzidou**[7], **Christoforos Nikolaides**[4,8], **Niki Savvidou**[9], **Savvas Katsouris**[10], **Maria Koliou**[11]

1 School of Clinical Medicine, University of Cambridge, Cambridge, United Kingdom, 2 The Roslin Institute and The Royal (Dick) School of Veterinary Studies, University of Edinburgh, Edinburgh, United Kingdom, 3 Infection Control Services, Nicosia General Hospital, Lemesou, Cyprus, 4 Cyprus Nurses and Midwives Association, Nicosia, Cyprus, 5 Intensive Care Unit, Nicosia General Hospital, Nicosia, Cyprus, 6 Pancyprian Medical Association, Nicosia, Cyprus, 7 Infection Control Services, Ammochostos General Hospital, Paralimni, Cyprus, 8 Infection Control Services, Limassol General Hospital, Nikaias, Kato Polemidia, Cyprus, 9 Infection Control Services, Paphos General Hospital, Paphos, Cyprus, 10 Infection Control Services, Larnaca General Hospital, Larnaca, Cyprus, 11 Department of Pediatrics, School of Medicine, University of Cyprus, Nicosia, Cyprus

* kk634@cam.ac.uk

## Abstract

### Introduction

The spread of COVID-19 into a global pandemic has negatively affected the mental health of frontline healthcare-workers. This study is a multi-centre, cross-sectional epidemiological study that uses nationwide data to assess the prevalence of stress, anxiety, depression and burnout among health care workers managing COVID-19 patients in Cyprus. The study also investigates the mechanism behind the manifestation of these pathologies, as to allow for the design of more effective protective measures.

### Methods

Data on the mental health status of the healthcare workers were collected from healthcare professionals from all over the nation, who worked directly with Covid patients. This was done via the use of 64-item, self-administered questionnaire, which was comprised of the DASS21 questionnaire, the Maslach Burnout Inventory and a number of original questions. Multivariable logistic regression models were used to investigate factors associated with each of the mental health measures.

### Results

The sample population was comprised of 381 healthcare professionals, out of which 72.7% were nursing staff, 12.9% were medical doctors and 14.4% belonged to other occupations. The prevalence of anxiety, stress and depression among the sample population were 28.6%, 18.11% and 15% respectively. The prevalence of burnout was 12.3%. This was in

**Data Availability Statement:** All relevant data are within the manuscript and its Supporting Information files.

**Funding:** The authors received no specific funding for this work.

**Competing interests:** The authors have declared that no competing interests exist.

parallel with several changes in the lives of the healthcare professionals, including; working longer hours, spending time in isolation and being separated from family.

## Discussion

This study indicates that the mental health of a significant portion of the nation's workforce is compromised and, therefore, highlights the need for an urgent intervention particularly since many countries, including Cyprus, are suffering a second wave of the pandemic. The identified risk factors should offer guidance for employers aiming to protect their frontline healthcare workers from the negative effects of the COVID-19 pandemic.

## 1. Introduction

Mental health constitutes one of the three main dimensions of overall health for individuals and communities alike. An important consideration is the mental health of healthcare workers (HCWs), as it can directly affect the quality of their work, and thereby the quality of patient care provided. According to the CDC, HCWs include all hospital workers that have been potentially exposed to infectious agents transmitted from other workers or patients [1]. The mental health of workers in general and health professionals in particular, may progressively deteriorate via prolonged exposure to a stressful working environment. Prolonged stress on doctors and other healthcare workers is tightly linked to an increase in job dissatisfaction, loss of professional performance, and decreased productivity [2]. This can have profound consequences both on workers' health and on the quality of care they provide. What is more, the experience of exhaustion by one worker may negatively impact everyone in their immediate working environment, creating a so called "domino effect". This is particularly prominent in medical settings that operate as large multidisciplinary teams such as the Intensive Care Unit (ICU), where the performance of HCWs is largely dependent on the performance of the team [3]. A higher percentage of emotional exhaustion among team members negatively impacts the interpersonal teamwork, and subsequently compromise patient care [4, 5]. Currently, the spread of the novel coronavirus has been deemed a major source of uncertainty, fear and anxiety for a lot of HCWs around the world, affecting their physical and psychological health in a number of ways [6]. The position of healthcare professionals in the frontline during the fight against the current pandemic requires physical and mental strength to overcome new and unknown challenges.

One axis along which mental health is routinely evaluated, is that of stress, anxiety and depression. That is because when these symptoms manifest in a physician or healthcare worker, they can have profound consequences on the quality of care provided. This can be due to a compromised patient-doctor relationship, or even medical errors [7]. Furthermore, prolonged experiences of anxiety and stress may also precipitate more severe pathological phenotypes like that of burnout; a state of physical and mental exhaustion. Among physicians, burnout is usually brought about by adverse working conditions including long hours, immense pressure and high levels of responsibility, which may either be exacerbated or prevented by other environmental factors [8]. Physician and nurse burnout has been repeatedly reported as a factor threatening patient care [9–13]. Making matters worse, recovery from burnout is not an easy process. The reported risk of burnout leads to a vicious feedback loop of causality with a decrease in performance, propagating the damage inflicted by stress on the medical worker [14, 15]. Therefore, the mental health status of all HCWs, at any given time, needs to be closely monitored, and any observed pathologies should be rapidly addressed.

The introduction of the novel coronavirus in late 2019 has been a major source of stress throughout the global population, with the medical community being one of the frontline sectors most severely affected [16, 17]. The association between a pandemic and the negative effects it may bring upon the mental health of HCWs is further supported by the literature exploring the effects of the 2003 SARS-CoV epidemic [18, 19], and the 2012 MERS-CoV epidemic [20]. Mental health stress may partly be attributed to an increased workload, caused by the large number of COVID-19 cases overwhelming the National healthcare systems, but it may also be attributed to other factors such as the fear of infection, stigmatisation, and loneliness [17]. Particularly prominent are the effects of the pandemic on the mental health of workers in the ICU, who are dealing with the most severely affected cases [21]. HCWs all over the world are currently experiencing a unique situation, with an immense load of responsibility resting on their shoulders. The professional performance of each HCWs could directly impact a country's immediate response to the pandemic, and the epidemiological trajectory that the virus will take as the compromised professional performance of workers in the COVID-19 hospital wards may propagate the risk of infection [22]. It is therefore essential that the mental state of the HCWs is closely monitored and any change for the worse is rapidly addressed. Key to offering a protective working environment of a country's HCWs is the understanding of mechanisms via which environmental "stressors" manifest into mental-health pathologies.

The response of the Cyprus government to the first wave of the COVID-19 pandemic, appeared to be an effective one, rapidly containing the spread of the virus and keeping the total number of deaths in the low double-digits. Central to this response, was a well laid out strategy adopted by the Nation's Ministry of Health, which made use of the entirety of its healthcare system [23], and the advantage of managing only a small population [24]. This prevented the national healthcare system from being overwhelmed during the first phase of the pandemic, by spreading out the load of work to institutions all over the country, but the impact of the pandemic on the mental health of the Cypriot HCWs remains to be determined. This article explores the degree of stress, anxiety, depression and burnout that HCWs experienced during the first wave of the pandemic. Furthermore, it investigates the impact of possible risk factors some of which could be reversed or managed in order to improve the wellbeing of professionals dealing with COVID-19 patients. As the pandemic continues to develop, these parameters should be taken under consideration in order to better prepare for the second and more intense wave of the pandemic.

## 2. Materials & methods

### 2.1 The study setting

This study took place in the Republic of Cyprus, during May and June 2020, well into the first wave of the COVID-19 pandemic. At the time, the number of cases per day was limited to under 20, with a total of four deaths being reported during the study period. However, the first wave had peaked in April, one month before the study took place, when there was 20–50 reported cases per day, and 15 total deaths [25]. The Healthcare system of Cyprus is divided into a public and a private sector. The nation's response to the pandemic was primarily handled by the public sector, with a few private healthcare professionals recruited to be part of the workforce. There are six major public hospitals; Nicosia, Larnaca, Limassol, Paphos and Ammochostos General Hospitals, alongside Makarios Hospital in Nicosia, and all six were directly involved in the response to the pandemic. The strategy adopted by the Ministry of Health in response to the pandemic goes as follows; In the first instance, suspected cases came into a specially modified section of the Emergency Department of their local public hospital. From there, patients were tested by rapid PCR test and confirmed cases which required

hospitalisation were pipelined into the reference hospital. The Ammochostos General Hospital was established as the reference hospital early on, and it was there that all confirmed cases of COVID-19 were treated. Patients who developed severe symptoms and needed intensive care, were moved to the ICU (Intensive Care Units) of Nicosia and Limassol General Hospitals, were resources were ample. This study takes place in the six major public hospitals of Cyprus, surveying their HCW involved in COVID-19 patient care.

## 2.2 The surveyed population

All HCWs working in dedicated hospital wards and intensive care units providing care to COVID-19 patients in Cyprus from May to June 2020 were eligible to participate in the study. Specifically, the sample population included staff at the six public hospitals of the country. All participants were, workers in either; (i) the emergency Department, (ii) the ward for suspected COVID cases, (iii) the COVID-ward or (iv) the ICU (Intensive Care Unit). Health professionals in the above-mentioned wards included physicians from different specialties, nurses, nurses' aides, physiotherapists, social carers and staff for cleaning and support services. The study was approved by the Cyprus National Bioethics Committee as well as by the Scientific Committee of the Agency for Public Hospitals.

## 2.3 Data collection

Data were collected using a 64-item web-based, anonymous, and self-administered questionnaire. The Infection Control nurses in each one of the public hospitals included, identified the HCWs that were working in wards with confirmed or suspected cases of COVID-19. The designated nurses were then given the questionnaire in an email form, as a link to the "Google forms" platform, and were instructed to distribute it to the HCWs. They were also given the option to request a paper copy. The emailed questionnaires were distributed to the HCWs work email accounts. The data were collected anonymously with no personal identifiable characteristics. No reminders were sent, and all responses included were collected from the first attempt. All forms completed on paper were collected by the infection control nurses, given back to the research team and were later transcribed electronically through Google forms to be integrated to the main database. The data collected online were downloaded as a Microsoft Excel spreadsheet.

## 2.4 The questionnaire

The questionnaire included a combination of original questions on demographics and work experience along with two internationally validated questionnaires; the depression, anxiety and stress scale (DASS-21) and the Maslach Burnout Inventory (MBI). The DASS-21 questionnaire was included to evaluate levels of depression, anxiety, and stress among health professionals. The DASS-21 is a composite 21-part questionnaire comprised of; (a) DASS-Depression, focusing on mood, motivation and self-esteem, (b) DASS-Anxiety, exploring occurrences of fear, panic and psychological-arousal and (c) DASS-Stress assessing tension and irritability [26]. DASS validity has been repeatedly confirmed over the years, as well as its applicability in healthcare [27], and non-healthcare [28] settings. The DASS-21 questionnaire is publicly available and freely used by researchers without the need to obtain a license [29]. Among the questions on the demographic profile of the participants, questions on height and weight were included, with the responses being used to calculate each individual's BMI (Body Mass Index).

Furthermore, the MBI (Maslach Burnout Inventory) was used to assess the levels of burnout among study participants. The MBI is comprised of 22 questions, and it is also a composite

questionnaire including five questions on depersonalisation (DP), eight questions on personal accomplishment (PA) and nine questions on emotional exhaustion (EE). The higher the participant scores on EE and DP, and the lower the score on PA, the higher the likelihood that they suffer from burnout. License to use the MBI questionnaire was purchased via Mind Garden ("*Remote Online Survey License*"). MBI is considered as one of the gold standards for burnout evaluation [30]. The MBI has been consistently used throughout the medical community, and it has been repeatedly validated as a measure of burnout among different populations [31]. However, there is no clear definition as to how burnout should be defined with the data obtained by the MBI. In particular, there is no consensus on which scores serve as thresholds to mark an 'abnormal' score in each parameter, and definitions of burnout diagnosis are based on either an abnormal score in all three parameters or a combination of one or two parameters [32]. In this study, we chose the most conservative approach, using the most commonly used thresholds as reviewed by Doulougeri et al. and defining burnout diagnosis as an abnormal score observed in all three parameters, which was suggest by Maslach, but only supported by a portion of the literature [32]. The questionnaire concluded with an open question, which read "Please describe in one sentence how the COVID19 pandemic has affected your life" giving study participants the opportunity to provide more detail on their personal experience of the effect of the pandemic. This questionnaire was pre-tested among the HCW of the research team and their feedback was followed to create the content of the final questionnaire.

## 2.5 Scoring

Responses to the DASS21 questionnaire were scored according to the scoring system described by the "*Manual for the Depression*, *Anxiety & Stress Scales*" [33]. Briefly, each answer was assigned a score from 0 to 3 and the final score per category was multiplied by 2. The recommended cut-off scores were used to classify each participant's response as normal, mild, moderate, severe and extremely severe for stress, anxiety and depression. To estimate prevalence each scale was recategorized into a binary variable, i.e., severe/extremely Severe vs normal to moderate. The scores from each of the six hospitals is displayed separately in order to evaluate whether the reference hospital of Ammochostos scored differently. To calculate the MBI, a score of 0–6 was assigned for each question and the scores were added up for each category. A cut-off of ≥27 indicated high Emotional Exhaustion, ≥10 indicated high depersonalization, and <34 indicated low personal accomplishment. These cut-offs were chosen based on guidance provided by similar studies exploring the prevalence of burnout in COVID-19 or ICU wards [32, 34–36]. Participants with high emotional exhaustion, high depersonalization and low personal accomplishment scores were classified as suffering from burnout. This is based on a statement by Maslach, the creator of the MBI, defining burnout as such, along with further support from the literature [37, 38].

## 2.6 Statistical analysis

Data were analysed using the R statistical software version 3.5.1 [39]. Package ggplot2 was used for plotting [40]. R package Performance Analytics [41] was used to visualise the relationship between DASS21 and MBI scores and estimate Spearman correlation coefficients for each pair. An exact binomial test was used to estimate confidence intervals of proportions using R package stats [39]. To investigate factors associated with low mental health and burnout four multivariable logistic regression models were built predicting the following outcome variables: 1) severe or extremely severe stress, 2) severe or extremely severe depression, 3) severe or extremely severe anxiety and 4) burnout. In order to select which variables to consider in the multivariable models univariable analysis was carried out. A linear model was used for

numerical predictors and fisher's exact test was used for binary/categorical predictors. Variables with a p value < 0.15 at the univariable analysis were considered for inclusion in the multivariable models [42]. Spearman correlation coefficients were estimated for each pair of variables and where high correlation was found between a pair, only one of the two variables were considered for inclusion in the model (the variable with lowest AICc in the univariable regression model). Variable selection was carried out using manual forward selection and the final model was chosen based on AICc a p-value >0.05. AICc was calculated using the MuMIn R package [43].

## 2.7 Qualitative analysis

The method of Inductive reasoning was used for the qualitative analysis of the responses to the open question, enquiring the ways in which the COVID-19 pandemic had impacted the lives of the participants. The analysis was initially performed by two researchers and was re-examined by two more independent researchers to confirm the reliability of the method. The responses were split into six thematic categories, depending on the conceptual interpretation of the content, with some responses often being placed in more than one category. The six categories were: (1) Reference to Stress, Anxiety, Depression or Burnout, (2) Infliction of negative emotions, (3) Infliction of positive emotions, (4) Experience of social isolation, (5) Reference to the fear of transmission, (6) Impact in an undefined way. *These categories emerged from thematic analysis based on the independent opinion of two of the co-authors.* Responses belonging in the latter category, were responses that clearly indicated the participant had been impacted in some way, but whether that was a positive or negative way was undefined (e.g. "I was affected very much"). Lastly, a small percentage of the responses (7/270) were offering no information on whether the participant was affected by the pandemic (e.g., one-word responses), and those were discarded. All data were collected and analysed in Greek, and all results presented are a translation in the English language performed by a bilingual researcher. Each translation was backtranslated in Greek to ensure the meaning was not altered.

## 3. Results

### 3.1 Demographic data

**3.1.1 The surveyed population.** A total of 381 HCWs completed the survey, from hospitals throughout the country. At the time, a total of 743 HCWs were working in the relevant wards and the breakdown of the response rate per hospital and overall is shown in S1 Table. Specifically, from Nicosia General (n = 75), Limassol General (n = 50), Larnaca General (n = 25), Paphos General (n = 67), Archbishop Makarios (n = 46) and Ammochostos General (n = 117), with the latter being the nation's "reference hospital" during the pandemic. The sample population appeared to be from a diverse demographic background (*Table 1*). Out of

Table 1. Demographic data of the sample population.

| Measure | Doctor | | Nurse | | Other healthcare professional | | Cleaning Staff |
|---|---|---|---|---|---|---|---|
| | Female | Male | Female | Male | Female | Male | Female |
| Sample Size | 28 | 21 | 202 | 75 | 18 | 7 | 30 |
| BMI mean (range) | 25.0 (18.2–61.7) | 26.5 (20.5–34.6) | 24.1 (17.2–64.9) | 25.8 (17.8–41.0) | 26.7 (17.1–38.0) | 22.4 (20.6–27.8) | 27.3 (18.1–37.5) |
| Voluntary participation | 6 (21.4%) | 8 (38.1%) | 36 (17.8%) | 8 (10.7%) | 3 (16.7%) | 1 (14.3%) | 8 (26.7%) |
| Increase in workload | 15 (53.6) | 13 (61.9%) | 103 (51%) | 36 (48%) | 9 (50%) | 1 (14.3%) | 16 (53.4%) |
| Change in daily duties | 25 (89.3%) | 19 (90.5%) | 174 (86%) | 56 (74.7%) | 14 (77.8%) | 6 (85.7%) | 28 (93.3%) |
| COVID-19 positive | 2 (7.14%) | 0 | 14 (6.93%) | 10 (13.3%) | 0 | 0 | 1 (3.3%) |

the total surveyed population (n = 381), 80% were female and 20% were male. Most of the respondents of the questionnaire worked as nurses (72.7%), with the rest being doctors (12.9%) or other healthcare professionals (6.6%) and cleaning staff (7.9%). A large percentage of the population (45.1%) reported an unhealthy BMI, defined as either underweight or overweight.

**3.1.2 Occupational profile of participants.** Most people surveyed (63%) had a long experience (>10 years) of working in their hospital, but most (84.5%) had seen a change in their daily duties in response to the pandemic. Alongside this change in duties, came a reported increase in their workload and a reduction in their time-off, with most participants (72.7%) taking a single day off during the week, and 6.3% even reporting 7-day work weeks. Together, this data suggests the establishment of a novel and unfamiliar working environment. Furthermore, a large portion of the surveyed HCW (81.6%) was not there by choice, but rather as a result of employer instructions. Out of those, 55.6% declared that the fear of potential infection had a strong negative impact on their mood at work. Adding to that, 17.9% of all participants declared that they did not feel safe by the protective measures taken and PPE (Personal Protective Equipment) offered by their hospitals. Moreover, 58% of participants had taken distancing measures from their families and loved ones, as 38.1% had moved to a separate house, with the remaining 19.9% remained in their house after subjecting it to "structural modifications".

## 3.2 DASS-21

**3.2.1 Components of the DASS-21.** The DASS-21 questionnaire was used to assess the mental health of the population using three distinct measures: Depression, Anxiety and Stress. The distribution of each score is shown in *Fig 1*. Each of the measures was considered in isolation, and findings indicated a relatively high prevalence of all three measures along the studied population. Specifically, 15.0% of the participants were classified as positive Depression, 28.6% were positive for Anxiety and 18.11% scored positive for stress. Numerical scores of all three measures were seen to be strongly correlated with one another (*S1 Fig*), indicating that it was the same portion of the population that scored highly across all three components. We then investigated the different factors associated with depression, anxiety and stress, and we divided them into two main categories: (i) occupational and (ii) non-occupational risk factors. Multivariable logistic regression models identifying factors associated with high levels of stress, anxiety and depression are shown in *Fig 2*.

**3.2.2 Impact on mental health by profession.** It became apparent from early on in our analysis, that the pandemic differentially affected the mental health of different healthcare professionals (*Table 2*). Most severely affected were the nursing and cleaning staff, which scored twice as high in prevalence of stress, anxiety and depression, as the medical doctors. For example, within the sample population of medical doctors, 8.16% scored "*severe/extremely severe*" levels of depression, 16.3% scored "*severe/extremely severe*" levels of anxiety and 10.2% scored "*severe/extremely severe*" levels of stress. This goes to contrast the sample population of the nursing staff, marking a prevalence of 17.3%, 32.1% and 20.9% in the measures of depression, anxiety and stress respectively. A similar prevalence was noted among the sample population of the cleaning staff. Lastly, a negative association was observed between level of education and levels of stress (OR = 0.08, p<0.05).

**3.2.3 Occupational risk factors.** Most of the factors identified were associated with an individual's working environment, and how that environment had changed during the pandemic. For example, people who reported an increase in their work hours were nearly twice as likely to experience higher levels of stress (OR = 2.02, p<0.05) and anxiety (OR = 1.65, p<0.05) than those who did not. Furthermore, the self-reported feeling of safety with the measures provided by each individual's hospital also exhibited a negative association with mental

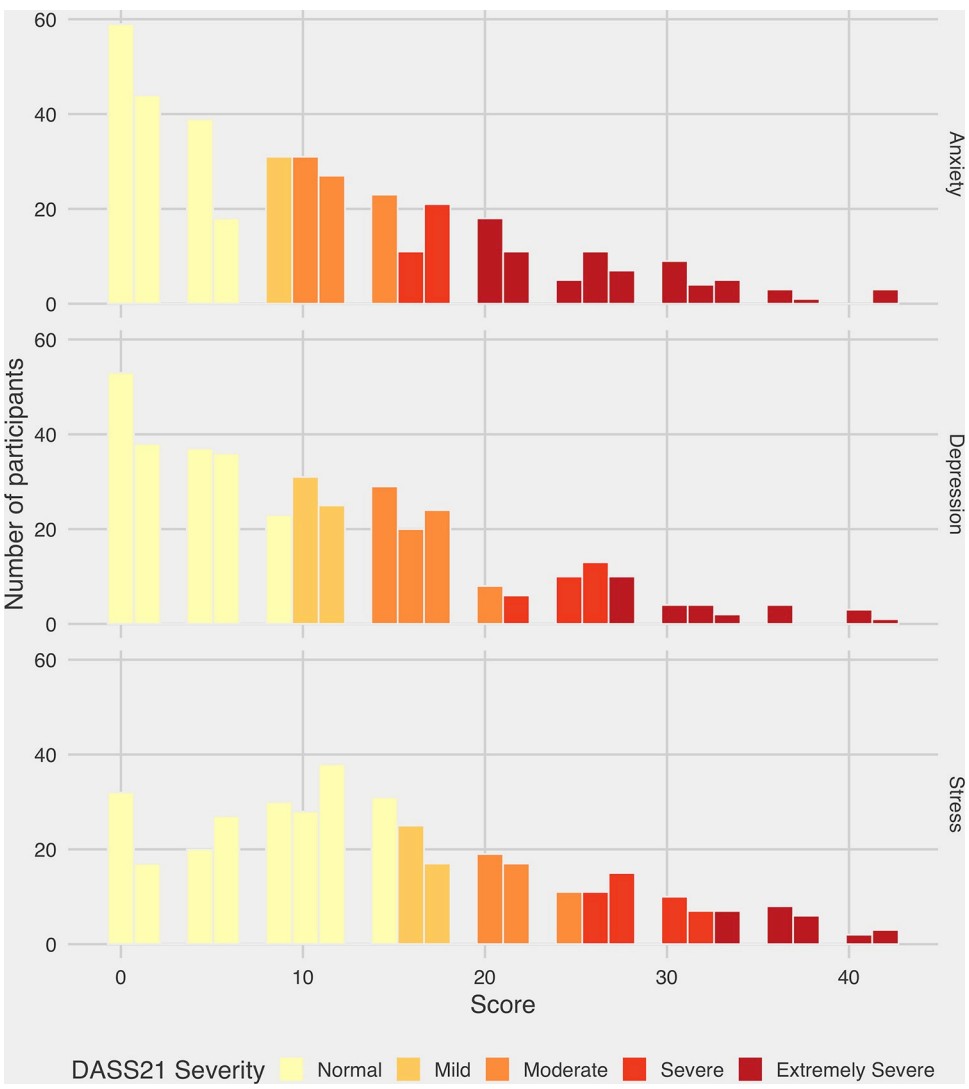

**Fig 1. Histograms showing the distribution of DASS-21 scores for each of the three measures; anxiety, depression and stress.**

health pathologies. People declaring an "*average*" or "*high/very high*" feeling of safety with the hospital measures were nearly half as likely to experience high levels of anxiety (OR = 0.43, p<0.05) and depression (OR = 0.42, p<0.05) than those that declared a "*low/very low*" feeling of safety. In addition, the central role of the individual's working environment was further highlighted by identifying a marked discrepancy between the mental health of people working in the Ammochostos General Hospital, when compared to all other healthcare centre. People at the Ammochostos General were seen to experience significantly less stress (OR = 0.07, p<0.05), anxiety (OR = 0.29, p<0.05), and depression (OR = 0.26, p<0.05), than people working in Nicosia General Hospital (the country's largest healthcare centre) or any other major hospital included in this study. What is more, participants from Ammochostos General reported a higher baseline feeling of safety with the hospital measures provided (*Fig 3*).

**3.2.4 Non-occupational risk factors.** The spread of the SARS-COV2 virus into a global pandemic has brought about a number of lifestyle changes for HCWs, that are not necessarily

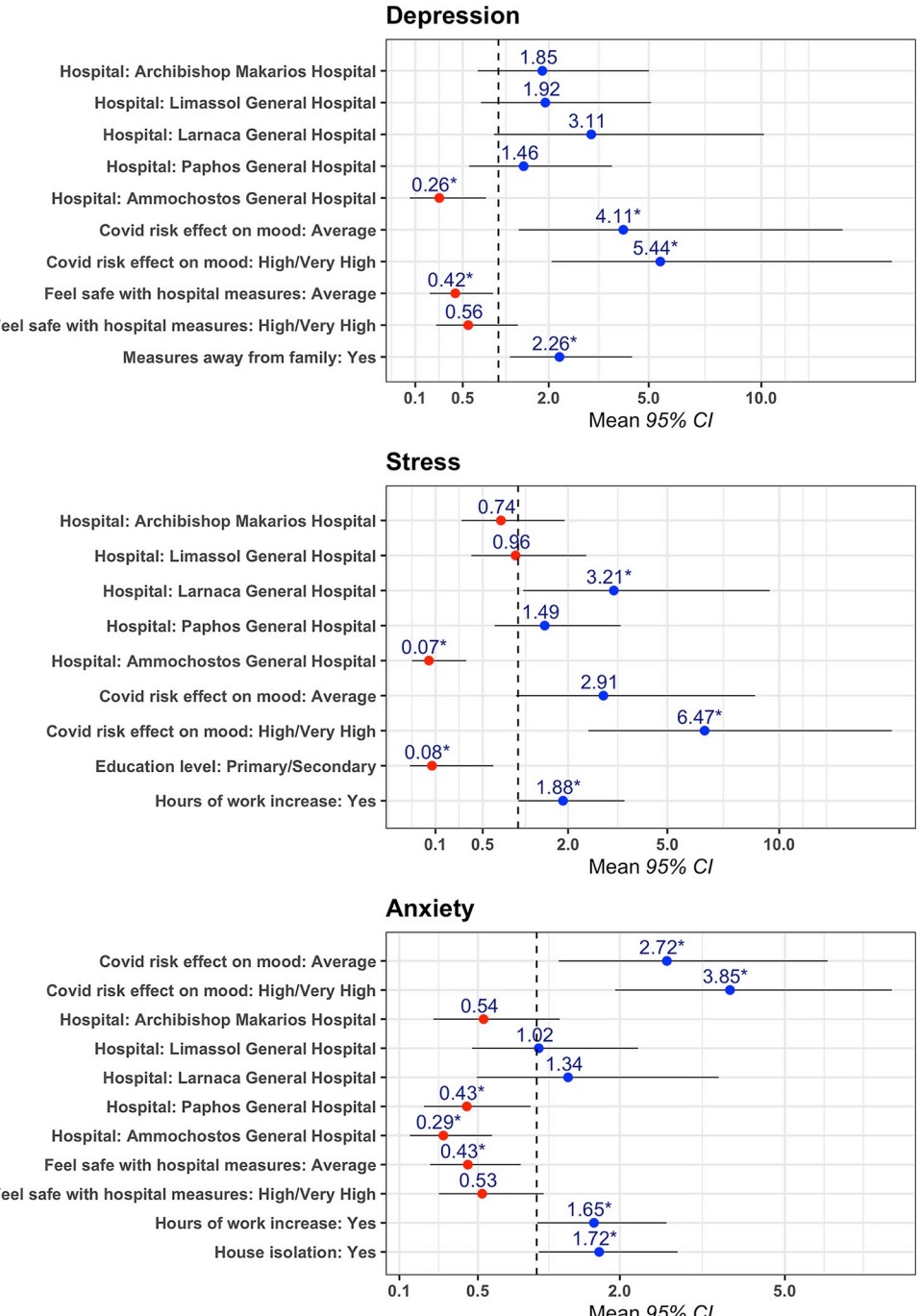

**Fig 2. Multivariable logistic regression models identifying factors associated with high levels of stress, anxiety and depression.**

part of an individual's working environment. For example, national guidelines on self-isolation following close contact with a confirmed case of COVID-19, compelled a number of healthcare professionals into taking measures away from their friends and family, which was reflected in increasing levels of anxiety and depression. Specifically, participants who had been subjected to house isolation were more likely to experience higher levels of anxiety (OR = 1.72, p<0.05),

**Table 2. Mental health by profession.**

| Occupation | DASS-21 | | | Maslach | | |
|---|---|---|---|---|---|---|
| | Anxiety | Depression | Stress | Depersonalisation | Emotional Exhaustion | Personal Accomplishment |
| Doctor | 7.3 (0–28) | 9.6 (0–28) | 12.2 (0–32) | 6.1 (0–24) | 26.0 (6–54) | 37.0 (24–48) |
| Nurse | 11.8 (0–42) | 11.4 (0–42) | 15.6 (0–42) | 8.4 (0–27) | 28.0 (0–54) | 35.5 (10–48) |
| Other Healthcare Professional | 6.3 (0–20) | 7.9 (0–20) | 11.2 (0–34) | 5.0 (0–22) | 24.8 (8–51) | 34.5 (17–45) |
| Cleaning Staff | 12.9 (0–42) | 11.4 (0–36) | 13.1 (0–36) | 6.7 (0–22) | 27.7 (7–49) | 32.8 (6–48) |

and participants that opted for taking distancing measures away from their families, experienced higher levels of depression (OR = 2.26, p<0.05). Furthermore, within our sample population, people that reported that the fear of a potential infection has a "*high/very high*" effect on their mood, were far more likely to experience high levels of depression (OR = 5.44, p<0.05), anxiety (OR = 3.85, p<0.05) and stress (OR = 6.47, p<0.05). Together, these data suggest that there are factors which span beyond the workplace of a healthcare professional, that can have a negative impact on their mental health. There were no associations between age or gender and either of the three pathologies measured.

### 3.3 MBI

**3.3.1 Burnout by profession.** The MBI (Maslach Burnout Inventory) is a three-part questionnaire that was used to assess the prevalence of burnout among the surveyed population. Each participant was allocated a score on each of the three constitutive measures; Emotional

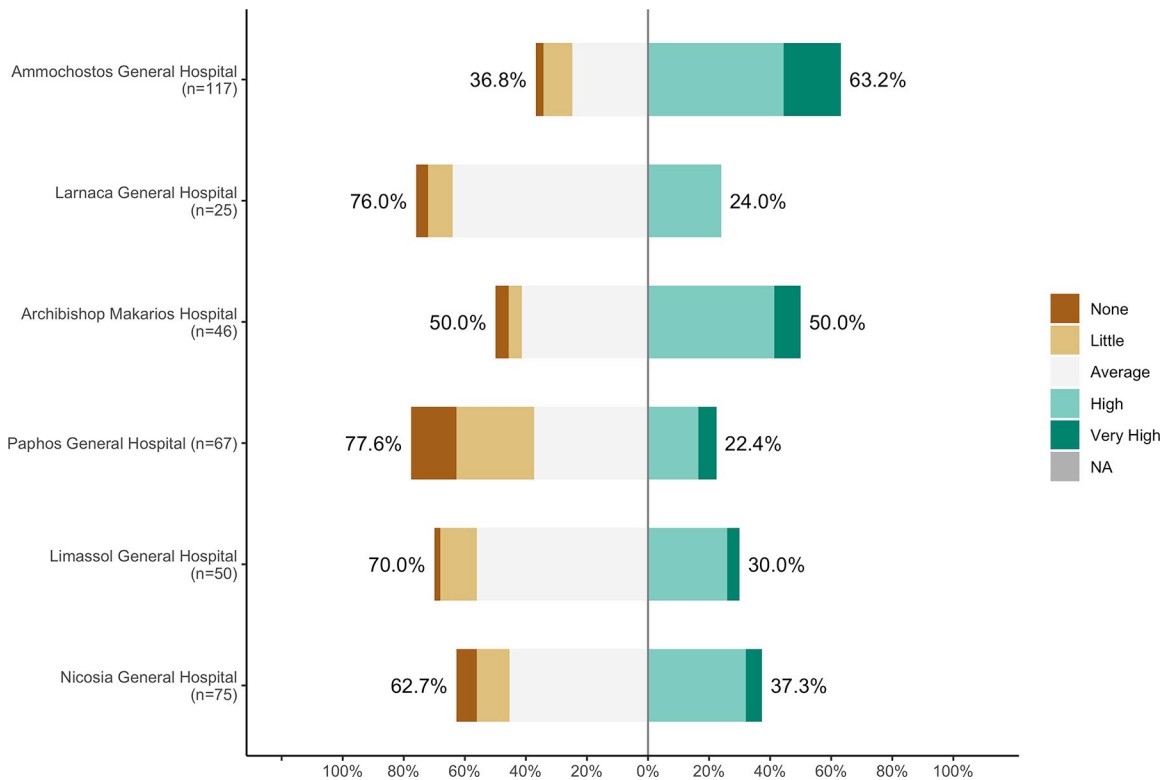

**Fig 3. Likert plot showing the extent to which workers of different hospital experience a feeling of safety with the measures taken by their employers.**

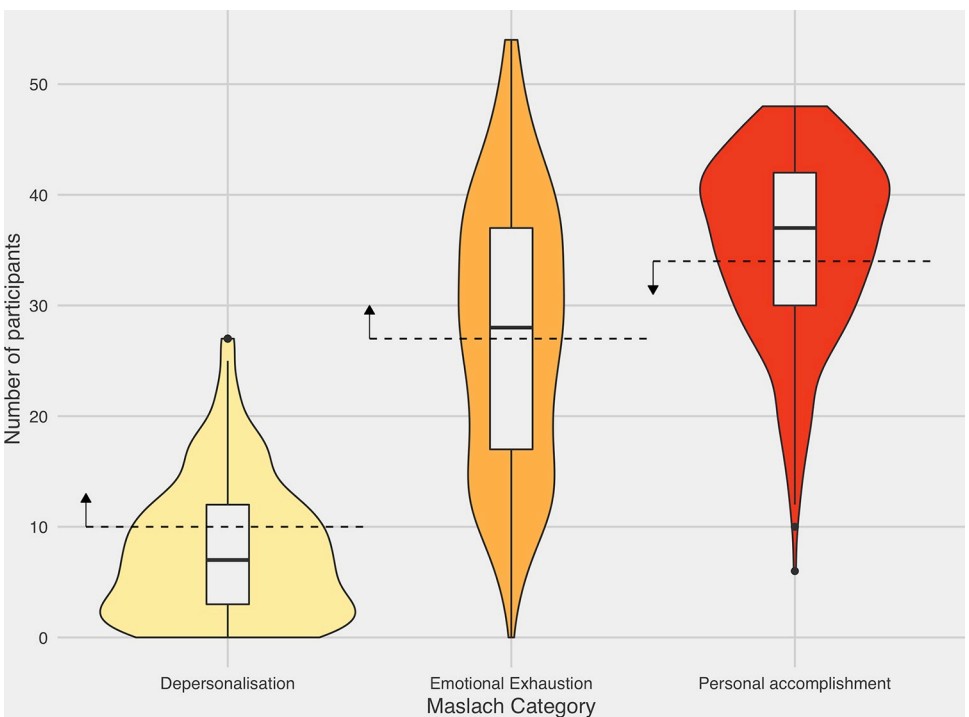

**Fig 4. Violin plot showing the distribution of MBI scores for each of the three measures of depersonalisation, emotional exhaustion and personal accomplishment.**

Exhaustion, Depersonalisation, and lack of Personal accomplishment (*Fig 4*) and a state of burnout was defined as a positive score in all three measures. Using this definition, the prevalence of burnout was estimated at 12.3%, among the surveyed population. When looking how the data varied between different professions, the nursing staff was again the group seen to be most severely affected, scoring a prevalence of 14.1%, as opposed to doctors (6.12%), other healthcare professionals (8%) and cleaning staff (10%). In an attempt to identify risk factors that promote a state of burnout, we explored the potential association between burnout and a set of selected variables.

**3.3.2 Risk factors for burnout.** Multivariable logistic regression-model analysis was performed to identify factors associated with a state of burnout (*Fig 5*). The effect of the pandemic on increasing rates of burnout seemed to be the result of a combination of occupational and non-occupational risk factors. To begin with, it was clear that people who reported that the fear of a potential infection had a strong impact on their mental state, were far more likely to burn out, than those who did not (OR = 3.12, $p < 0.05$). Furthermore, people that were compelled into isolation following exposure to the virus, were also more likely to experience burnout than the people who did not have to undergo a self-isolation (OR = 2.19, $p < 0.05$). Moreover, it seemed to be the case that people with an unhealthy BMI were also more likely to experience burnout. Lastly, there was a strong negative association between the feeling of safety with the protective measures taken by the hospital, with the experience of burnout. There was no association between burnout and age or gender.

## 3.4 Qualitative analysis

**3.4.1 Analysis of open questions.** Participants were asked to respond, in one sentence, how they were affected by the pandemic. This was an open-ended question, which was

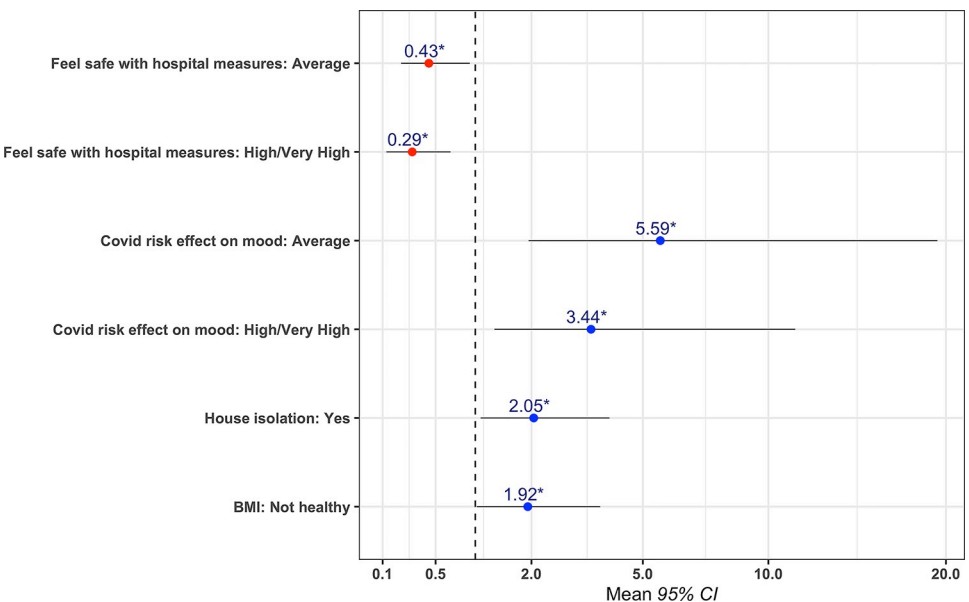

**Fig 5. Multivariable logistic regression model identifying factors associated with high levels of burnout.**

completed by 277 out of the 381 participants, out of which 7 responses were discarded due to lack of sufficient information. A summary of the data can be found on *Table 3*. A more detailed description of the qualitative data collected, alongside examples, can be found in the (S1 Table). 83 responses (30.7%) were classified as 'Reference to Stress/Anxiety/Depression or Burnout', due to their reference to these states of mental illness. Examples included; *"[I experi-enced] increased levels of stress, loneliness, worse mood and emotional exhaustion"*, and *"I still feel exhausted, in both my body and mind"*. Furthermore, 71 responses (26.3%) were classified as 'impacted the person in an undefined way', as they revealed little information regarding the mood/mental health of the individual. A representative example was *"I was affected very much"*. Then, 41 responses (15.2%) fell in the category 'inflicted positive emotions', as they dealt with themes of; optimism, resilience, offering, self-esteem, mutual respect and joy. Exam-ples of such responses were; *"The pandemic has made me a better professional"*, or *"[the pan-demic] had a positive effect on me, allowing me to mature in multiple aspects of my character"*. In a similar way, 37 responses (13.7%) were placed in the category of 'inflicted negative emo-tions', as they revealed the experience of a negative emotion. *"It negatively affected my daily life"*, acts as an example of such responses. Moreover, 17 responses (6.3%) discussed the theme

**Table 3. Results obtained from the qualitative analysis of how the COVID-19 pandemic has impacted the sample population.**

| Qualitative Analysis | |
|---|---|
| **Theme** | **Number (%):** |
| 1. Reference to Stress/ Anxiety/ Depression/ Burnout | 83 (30.7%) |
| 2. Impact in an undefined way | 71 (26.3%) |
| 3. Inflicted positive emotions | 41 (15.2%) |
| 4. Inflicted negative emotions | 37 (13.7%) |
| 5. Social Isolation | 17 (6.3%) |
| 6. Impacted by fear of transmission | 12 (4.4%) |
| **Total number of responses** | **270** |

of social isolation and its negative effects. Examples include: "*It affected the social lives of both myself and my family*", and "*I miss my kids and my husband*". Lastly, 12 responses (4.4%) were discussing the fear of transmission. For example, "*[I feel] scared when I come in contact with a confirmed case of Covid, both for myself and for my family*".

## 4. Discussion

This is a multi-centre, cross-sectional, epidemiological study that has collected nation-wide data from a large sample of HCWs directly involved with the management of COVID-19 patients. The sample population included HCWs from a range of different professionals, the most prevalent being nurses (73%) and doctors (13%). This study marks the first investigation in the levels of stress, anxiety, depression and burnout among first-line HCWs during the COVID19 pandemic in Cyprus. In addition, it also sheds light onto a number of underlying causes that might contribute to the manifestation of these pathologies.

The occupational profile of the sample population revealed some insights into the changes in the workplace provoked by the pandemic. For example, the workload of most healthcare workers had increased, which corresponded to a decrease in their weekly days off. In addition, a large proportion of the surveyed population declared being in the frontline following employer's instructions, rather their own will. Given that direct contact with COVID-19 patients has been repeatedly reported as a risk factor for emotional distress, this effect would be exacerbated in a setting where the contact is not voluntary [44–46]. Lastly, a significant proportion of the population appears to feel unsafe in their working environment despite the use of PPE. Reports throughout the world support the importance of PPE and material equipment in providing a feeling of safety among healthcare workers [47]. Together these changes appear to contribute towards creating a more hostile and stressful environment, that can take a toll on a professional's mental health. Furthermore, the pandemic appeared to also bring about changes in the workers' life, outside the workspace. This is primarily due to the need for implementation of social distancing measures away from family. Separation from loved ones was the reason many participants declared experiencing negative emotions, when asked how their lives were impacted by the pandemic. This is a link that has been reported by healthcare professionals all over the world, highlighting the importance of social inclusion and support from family and friends [48, 49]. All in all, these data suggest that when trying to appreciate how the lives of the HCWs were changed during the pandemic, it is important to look past the workplace and get a holistic view of the person's environment.

A relatively high prevalence of the three parameters examined by DASS 21 was observed in the sample population: Anxiety (29%), Stress (18%) and Depression (15%). There is limited information regarding the prevalence of this pathologies among the target population prior to the pandemic, but the values of the collected data follow a trend that was observed all over the world during the first wave of the pandemic [50, 51]. Furthermore, a strong correlation was observed between all three parameters, highlighting the extent to which the mental health of certain individuals was compromised, and the urgent need for an intervention. This is further supported by the emotional statements declared by some participants (*section 3.4.1*), describing their debilitating experience of stress and anxiety. Moreover, the pandemic appears to have a differential impact in HCWs of different professions, most strongly affecting the nursing staff. This pattern is in agreement with previous reports on the mental health of HCWs workers during the SARS epidemic, throughout the globe [49, 52–54]. In addition, a differential effect was observed between HCWs working in different hospitals, with people in Ammochostos General hospital (AGH) consistently scoring lower levels of all stress, anxiety and depression, and reporting a stronger feeling of safety towards the protective measures taken by their

hospital. This suggests the exercise of a particularly positive practice at AGH which managed to minimise any negative effects of the pandemic on its workers' mental health. A potential explanation would be that AGH was the nation's "reference hospital" in the response to the pandemic, and consequently, it was the institution that received the highest levels of preparatory material aid. Lastly, there were a number of other potential risk factors observed to contribute in the exacerbation of stress, anxiety and depression. Namely, longer working hours, fear of infection, experience of in-house isolation and separation from family. This highlights how the introduction of this novel coronavirus into people's lives was associated with a potential threat to their physical health, which acted in parallel with their distancing from society, to exacerbate any negative experiences of stress, anxiety and depression.

In a similar way, burnout was detected in a significant portion of the sample population (12.3%). This is despite using the most conservative definition of burnout using the data obtained from MBI. The most recent study that includes an estimate of the prevalence of burnout in a relevant population was conducted by Raftopoulos et al. 2012 [55], among Cypriot nurses. They estimated a prevalence of 12.8%, compared to a value of 14.1% calculated by this study, among the sample population of nurses. Furthermore, Raftopoulos et al. used a more lenient definition of burnout. For these reasons, we believe that we have seen an increase in the prevalence of burnout between 2012 and now. Furthermore, in our sample population, there was little correlation observed between the measure of burnout and scores on the three parameters of DASS-21 (stress, anxiety and depression), highlighting that the different portions of the population were affected in different ways. Differential experience of burnout was observed among HCWs of different professions, with the nursing staff reporting twice as high levels as the doctors. Potential risk factors contributing to the manifestation of burnout included fear of infection, experience of in-house isolation and an unhealthy BMI. Significant overlap between the identified risk factors for burnout, and risk factors for depression, anxiety and stress, imply that the pandemic can impact different aspects of the population's mental health via a conserved mechanism. This is supported further by the qualitative data obtained from the questionnaire, as several people reported social isolation and the fear of the infection to be a contributing factor to their mental health deterioration.

There were several strengths and limitations presented by the methodology used in this study. For example, a notable strength is the large sample size, collected across multiple centres throughout the nation. This allowed for a representative sample population, strengthening the reliability of the data collected. Furthermore, the use of two validated questionnaires in the collection of data regarding stress, anxiety, depression and burnout ensured a high validity of the experimental model. In addition, the use of an open question at the end of the questionnaire, offered an opportunity for participants to express concerns not necessarily surveyed for in the other close-ended questions. This opportunity was utilised by a large portion of the sample population (~72%) and offered a deeper insight into their experience which complemented well the quantitative data collected. Lastly, this study offers an insight into how small countries were able to manage the first wave of the pandemic. Which such countries may have the benefit of easily controlling an epidemic due to the small population, the often limited resources in equipment, trained personnel and or expertise may result in higher pressure in HCWs and the health care system in general. This study reveals the potential consequences of working with limited resources during the early stages of a new pandemic, on the mental health of the HCWs in a small country like Cyprus. However, several limitations were also presented in the process of data collection. For example, data were collected through the voluntary completion of a questionnaire, which implies that it would be mostly people with strong feelings towards the pandemic that would be concerned with completing the document. However, both questionnaires included (DASS-21 and MBI) were validated in the international population, and

they collectively comprise the a reliable approach to assessing the mental health of healthcare professionals [27, 31, 56, 57]. The questionnaires were not validated for the Cyprus population in the past. Moreover, there were limitations presented during the data analysis. As mentioned, there is not a single definition of burnout, and there are several approaches to interpretation of the data collected from the MBI questionnaire [32]. Our decision was to follow the most conservative approach to constructing the definition of burnout, as we found that the alternatives tend to represent an overestimation of the true prevalence. It is important to remember to take into consideration when comparing our results with previous or future studies.

In conclusion, this study has revealed a number of potential risk factors contributing to the experience of stress, anxiety, depression and burnout. The next step is using these data to construct a number of targeted hospital policies that will protect HCWs. For example, redistributing work hours such that there are a few intense work weeks followed by a few weeks off will permit HCWs to come in closer contact with their families and will minimise time spent in isolation. This can also be achieved by offering of frequent testing, as a negative test will reassure a low risk of transmission and will permit contact with family. Furthermore, in cases where isolation is necessary and separation from family is required, social support should be offered to the affected individual such that the negative effects of this experience are limited. Moreover, the provision of effective PPE, alongside with comprehensive instructions and training on how to use it are paramount for ensuring HCW safety. This can also promote a higher level of trust experienced by HCWs towards their institution. It is important to acknowledge the negative effects that the pandemic has brought about on the frontline healthcare workers of this nation and promote policies to protect them. Insights provided by the current study provide a useful framework for employers to minimise negative effects of the pandemic.

## Supporting information

**S1 Fig. Correlation between each of the scores obtained from all three measures of DASS-21 (stress, anxiety and depression), and all three measures of MBI (emotional exhaustion, depersonalisation and personal accomplishment).**
(TIF)

**S1 Table. Number of respondents and total number of HCWs eligible to take part in the study at each of the 6 hospitals and response rates.**
(DOCX)

## Acknowledgments

The authors would like to express their gratitude towards all the frontline workers who took the time to complete our questionnaire and reflect on their emotions during the first wave of the COVID19 pandemic.

## Author Contributions

**Conceptualization:** Stella Mazeri, Despo Constantinou, Anna Vavlitou, Marios Karaiskakis, Maria Koliou.

**Formal analysis:** Despo Constantinou.

**Investigation:** Demetra Kourouzidou, Christoforos Nikolaides, Niki Savvidou, Savvas Katsouris.

**Methodology:** Stella Mazeri, Anna Vavlitou, Maria Koliou.

**Project administration:** Anna Vavlitou.

**Resources:** Marios Karaiskakis, Maria Koliou.

**Supervision:** Anna Vavlitou, Maria Koliou.

**Writing – original draft:** Konstantinos Kapetanos.

**Writing – review & editing:** Konstantinos Kapetanos, Stella Mazeri, Despo Constantinou, Anna Vavlitou, Marios Karaiskakis, Maria Koliou.

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
