## [Decision Letter · Decision Letter 0]

29 Jun 2021

PONE-D-21-03042

Exploring the Factors Associated with the Mental Health of Frontline Healthcare Workers during the COVID19 pandemic in Cyprus

PLOS ONE

Dear Dr. Kapetanos,

Thank you for submitting your manuscript to PLOS ONE. After careful consideration, we feel that it has merit but does not fully meet PLOS ONE’s publication criteria as it currently stands. Therefore, we invite you to submit a revised version of the manuscript that addresses the points raised during the review process.

Please ensure that you carefully address all of the points raised by the reviewers when preparing your revisions. Please pay particular attention to explaining the aspects of your study design and methods that the reviewers have indicated need further clarification.

We look forward to receiving your revised manuscript.

Kind regards,

Jamie Males

Staff Editor

PLOS ONE

Journal Requirements:

2. In your Methods section, please provide a justification for the sample size used in your study, including any relevant power calculations (if applicable).

Please provide additional details regarding participant consent. In the ethics statement in the Methods and online submission information, please ensure that you have specified (1) whether consent was suitably informed and (2) what type you obtained (for instance, written or verbal). If your study included minors under age 18, state whether you obtained consent from parents or guardians. If the need for consent was waived by the ethics committee, please include this information.

In Table 1 please clarify how BMI was  measured for all participants, and please provide a justification in the Methods section regarding the selection of BMI as a study variable. Should this variable be reported as the 'MBI score'  please correct the table accordingly.

Reviewers' comments:

Reviewer's Responses to Questions

**Comments to the Author**

1. Is the manuscript technically sound, and do the data support the conclusions?

Reviewer #1: Partly

Reviewer #2: Yes

2. Has the statistical analysis been performed appropriately and rigorously? 

Reviewer #1: Yes

Reviewer #2: Yes

3. Have the authors made all data underlying the findings in their manuscript fully available?

Reviewer #1: Yes

Reviewer #2: Yes

4. Is the manuscript presented in an intelligible fashion and written in standard English?

Reviewer #1: Yes

Reviewer #2: No

5. Review Comments to the Author

Reviewer #1: Thank you for your submission. I believe the topic is very timely and interesting to the readers of PLOS ONE. However, there are few points to be considered revision.

[major revision]

1. It is not very clear why you needed to display the data separately for each six facilities. It gives an impression that this is not the common results that can be applied to other facilities. You may consider only displaying the results of all facilities together or state in the method/results why it need to be displayed separately.

2. I believe the significance of your manuscript is the mixed method (using both quantitative and qualitative data). and the findings from the qualitative data were very impressive. Would you consider discussing deeper regrading your findings? (ie., This qualitative data could explain why people does/doesn't burn out.)

[minor revision]

1. in line 176, change "Maslach Burnout Inventory (MBI)" to "MBI", since you have already explained the acronym in line 170.

2. tables and figures are not very easy to read. you may consider adding vertical lines appropriately.

3. for table 3, you may add the percentage of each numbers (not just in the text), so the reader can understand the weight of each theme.

Reviewer #2: Dear authors,

Firstly, I would like to congratulate you for the research you carried out, as it focus on a relevant topic and seems to have have been carried out with rigor. However, there are some issues you must address:

1) The use of the English language is not bad, but there are some typographical errors along the manuscript. Thus, I think your paper must be proofread by a native professional translator.

2) Although your paper presents some interesting information and findings, it is extremely long, so, it is not "reader-friendly". I recommend you keep the essential information in the manuscript and present the additional information in supplementary files.

3) A lot of research has already been conducted on this topic and, indeed, your paper does not add new knowledge. It just presents national data but you should not forget that PLOS ONE is a journal which is read by an international audience. Thus, you must make clear what your paper adds to the body of knowledge on that domain, and why it is relevant for an international audience.

4) You stated that data collection was carried out in several hospitals and that your sample was composed by 381 healthcare workers. But, to how many healthcare workers did you send the questionnaire? I mean, what was the response rate? That information is quite important and, if there is a low response rate, that can be a significant limitation of your study.

5) You included in your sample cleaning staff. Why did you consider cleaning staff as healthcare workers? That is a bit strange. If you really want to keep that staff in your sample, you should present a clear definition of "healthcare workers" in which it would make sense to include cleaning staff.

6) You use the term "frontline healthcare worker" several times along the manuscript. If you really want to use that term, you must clearly define what is a "frontline healthcare worker". However, that is not really a good term, as it is much more journalistic than scientific.

7) You presented an explanation of Cyprus response to the pandemic. However, you did not present the pandemic situation in Cyprus during the period of data collection, and that would be quite relevant. How many cases per day? How many hospitalizations per day? How many deaths per day? Was data collection carried out in the peak of a covid-19 wave or in a more stable period?

8) You must specify a bit more the data collection procedures. Who did send the questionnaires to the potential participants? The research team? How did you access the e-mail address of the participants? Did you send reminders? How many? Or was the questionnaire sent by the hospital? How may those options have influenced your results? How did you collect paper versions of the questionnaire? Was not Cyprus in lockdown during that period?

9) Were the assessment tools (MBI and DASS21) validated for the Cyprus population? If yes, you should present the most relevant psychometric properties of those tools in its version validated for the Cyprus population.

10) Did you pretest the questionnaire, considering there were some question which were developed by the research team? If yes, with ho many participants? What was their feedback on the data collection tool?

11) You stated that (line 218): "Variables with a p value < 0.15 at the univariable analysis were considered foe inclusion in the multivariable models". Did you mean "p value < 0.05". Or, if you really mean p < 0.15, why that option (which is not quite usual)?

12) In line 230 you stated that "the responses were split into six thematic categories, depending on the conceptual interpretation of the content". Were those categories predefined or they emerged from the thematic analysis?

13) In line 472 you stated that "a notable strength is the large sample size, collected across multiple centres throughout the nation. This allowed for a representative sample population". However, having a large sample (and your sample, indeed, is not that "large") does not guarantee a sample is representative. How can you assume that your sample is representative?

6. PLOS authors have the option to publish the peer review history of their article (what does this mean?). If published, this will include your full peer review and any attached files.

Reviewer #1: No

Reviewer #2: No

---

## [Author Response · Author response to Decision Letter 0]

15 Aug 2021

Thank you very much for your consideration and your feedback. Please find all our comments included in the "Response to reviewers" letter, that has been attached alongside the manuscript.

---

## [Decision Letter · Decision Letter 1]

29 Sep 2021

Exploring the Factors Associated with the Mental Health of Frontline Healthcare Workers during the COVID19 pandemic in Cyprus

PONE-D-21-03042R1

Dear Dr. Kapetanos,

We’re pleased to inform you that your manuscript has been judged scientifically suitable for publication and will be formally accepted for publication once it meets all outstanding technical requirements.

Kind regards,

Francisco Sampaio, Ph.D.

Guest Editor

PLOS ONE

Additional Editor Comments (optional):

Dear authors,

Your paper was revised once again by two independent reviewers. According to their opinion, all their previous comments were successfully addressed, so your paper can now be accepted for publication. Congratulations for that!

Reviewers' comments:

Reviewer's Responses to Questions

**Comments to the Author**

1. If the authors have adequately addressed your comments raised in a previous round of review and you feel that this manuscript is now acceptable for publication, you may indicate that here to bypass the “Comments to the Author” section, enter your conflict of interest statement in the “Confidential to Editor” section, and submit your "Accept" recommendation.

Reviewer #1: All comments have been addressed

2. Is the manuscript technically sound, and do the data support the conclusions?

Reviewer #1: Yes

3. Has the statistical analysis been performed appropriately and rigorously? 

Reviewer #1: Yes

4. Have the authors made all data underlying the findings in their manuscript fully available?

Reviewer #1: Yes

5. Is the manuscript presented in an intelligible fashion and written in standard English?

Reviewer #1: Yes

6. Review Comments to the Author

Reviewer #1: Thank you for revising the manuscript following reviewers suggestions. The significance of your manuscript which is the mixed method (using both quantitative and qualitative data) are strengthened and I believe your manuscript is now ready for publication. Thank you for submitting such a timely and inspiring manuscript. I believe your findings would help many healthcare practitioners around the world.

7. PLOS authors have the option to publish the peer review history of their article (what does this mean?). If published, this will include your full peer review and any attached files.

Reviewer #1: No

---

## [Editor Report · Acceptance letter]

5 Oct 2021

PONE-D-21-03042R1 

Exploring the Factors Associated with the Mental Health of Frontline Healthcare Workers during the COVID19 pandemic in Cyprus 

Dear Dr. Kapetanos:

I'm pleased to inform you that your manuscript has been deemed suitable for publication in PLOS ONE. Congratulations! Your manuscript is now with our production department. 

Kind regards, 

on behalf of

Professor Francisco Sampaio 

Guest Editor

PLOS ONE